# A Semantic Data Augmentation driven Nested Adapter for Video Moment Retrieval

## Abstract

Existing transformer-based video-moment retrieval models achieve sub-optimal performance when using the pretrain-finetuning learning paradigm – a pretrained multimodal encoder is finetuned using the target training data. While current work has explored different model architectures and training paradigms to explore this problem, the problem of data dilemma has been under addressed. Specifically, there exists high diversity of how semantic is captured in textual query and the training dataset only consist of limited moment-query pairs for the highly diverse moments. This work addresses this problem with a novel nested adaptor and a LLM-driven semantic data generation pipeline. First, a LLM-driven data augmentation generates queries that are semantically similar to the ground truth, which enrich the semantic boundary captured by textual query. We empirically analyze the effectiveness of data augmentation, and proposed a simple yet effective quality measure to retain high quality samples. Second, we propose a novel nested adapter that utilises both augmented queries and human annotated queries for model coarse-tuning and fine-tuning, respectively. By combining semantic perturbation with domain adaptation, our approach addresses the variability in video content while capturing nuanced features more effectively. Experimental results on various baseline models show the efficacy of our proposed approach.

## 1 Introduction

The proliferation of digital devices and platforms has led to a significant increase in the quantity and variety of videos. Due to its inherent complexity and richer semantic information compared to images and text, the demand for video understanding is on the rise Zhang et al. (2023). Recently, the novel and challenging tasks of video moment retrieval (MR) and highlight detection (HD) Lei et al. (2021) have attracted significant attention Ma et al. (2023); Yu et al. (2024); Xu et al. (2023); Lin et al. (2023a); Jang et al. (2023); Moon et al. (2023b;a). These tasks involve localizing specific moments within videos and determining the corresponding importance scores based on user-provided queries. An inherent challenge in these tasks is the need to efficiently retrieve desired moments from a vast array of videos and to navigate videos according to user preferences effectively. Additionally, it requires not only capturing temporal intra-modal context but also aligned cross-modal interaction of visual and textual features which makes it more challenging and non-trivial.

Recent studies Yan et al. (2020); Ling et al. (2022); Zhang et al. (2022) in image retrieval tasks have achieved efficient and effective retrieval of specific items from a large dataset. These approach generally utilizes a pretrain-finetune paradigm Liu et al. (2021a); Tang et al. (2024) – pre-training models on large datasets followed by fine-tuning on task-specific data – which has become a benchmark in training high-performance models. Initially popularized by architectures like ResNet He et al. (2016), this approach has continued to yield impressive results with the advent of visual transformers, such as ViT Liu et al. (2023); Khan et al. (2022); Tan et al. (2021); Bazi et al. (2021). The visual transformers leverage self-attention mechanisms to capture global dependencies and relationships within images, allowing them to effectively encode complex visual information. Notably, even with limited training data on target task, visual transformers have shown strong performance for image classification and retrieval tasks Lanchantin et al. (2021); Song et al. (2023), indicating their robustness and generalization ability across different datasets and tasks.

However, applying the pretrain-finetune paradigm for video-based moment retrieval tasks faces significant challenges Selva et al. (2023); Radford et al. (2021). The complexity of video data, as well as the rich and diverse natural language query, makes it difficult to train a discriminative model using conventional methods. *We argue that one underestimated factor in training model for video MR and HD lies in the data dilemma*. First, there's a substantial *domain gap* between the pre-training dataset and the target dataset. For example, the performance of a CLIP pretrained model is 70.97% lower than current state-of-the-art Moon et al. (2023b) for mAP@0.5. Second, the existing training dataset of video MR tasks is limited, where the available videos are sparse w.r.t. the potential diversity of unseen videos. This can lead to a reduction in the discriminative power of the pretrained model during fine-tuning stage Jaiswal et al. (2024). Third, each moment is labelled with a single natural language query, yet, there exists multiple variations of how semantic is captured in textual query.

To address the challenges inherent in video MR, we propose a novel approach that leverages both data augmentation and adaptive learning techniques to develop better aligned encoders. A central difficulty in this task arises from the rich and diverse content in videos, where a single segment can be described in multiple ways, depending on the focus of the caption. A single query often fails to capture the full semantic diversity of the content. To overcome this limitation, we employ data augmentation with large language models (LLMs) to generate varied, semantically coherent rephrasings of queries. This process expands the diversity of the training set and enables the model to generalize more effectively across different formulations of the same query, enhancing retrieval performance.

In conjunction with data augmentation, we utilize learnable adapters to enhance both *fine-tuning* and *coarse-tuning*. The standard fine-tuning approach adapts the model to the target dataset based on the original queries. In contrast, coarse-tuning involves using Nested Adapters (Sec. 3.4), which are trained with augmented queries, allowing the model to better capture the broad semantic space of queries while mitigating the effect of noisy augmentations. Our approach is model-agnostic, i.e., it can be applied across various architectures without being tied to a specific model, further enhancing its adaptability. This tuning strategy enables the model to retain its discriminative power while making it more adaptable to the diverse characteristics of the video data. By integrating both data augmentation and adapter-based learning, we aim to enhance the model's generalization capability, making it more effective in real-world moment retrieval applications. We identify that the core challenge of the MR task lies in the data dilemma, specifically (1) the domain gap between the pretrained model and the target dataset, (2) the sparse and limited availability of fine-tuning samples, and (3) the reliance on a single natural language query, which limits the generalizability of the fine-tuned model.

Our key contributions in this work are as follows:

- To address the aforementioned challenges, we introduce a novel integration of LLM-based data augmentation and learnable adapters for domain adaptation. Our approach combines LLM-driven query diversity with cross-modal refinement, using adapters to bridge the domain gap and enrich video representations.

- Specifically, we utilize data augmentation to diversify the training set and improve generalization capabilities of the model. In addition, we propose a Nested Adapter that uses semantic data augmentation for additional model coarse-tuning during the model fine-tuning process.

- Through extensive experiments and ablation studies, we empirically demonstrate the effectiveness of our model-agnostic approach, which can be applied across a variety of baseline architectures, resulting in improved performance across multiple MR baselines without being tied to any specific model.

## 2 RELATED WORK

### 2.1 VISUAL UNDERSTANDING OF FOUNDATION MODELS

The evolution of foundation models, as outlined in studies such as Li et al. (2022), has indeed led to significant advancements in various tasks within computer vision and natural language processing. However, while these models have demonstrated remarkable performance on single modalities like

text or images, they often struggle with understanding videos and handling multiple modalities effectively, as noted in studies such as Bazi et al. (2021); Song et al. (2023); Radford et al. (2021); Li et al. (2023).

Foundation large language models (LLMs) such as GPT (Generative Pre-trained Transformer), as highlighted in Shen et al. (2024), are primarily designed for tasks related to language generation, comprehension, and text classification. However, MR involves retrieving specific segments or moments from multimedia data, particularly videos, based on natural language queries. This task requires understanding both the textual queries and the visual content of the videos, which may be beyond the capabilities of LLMs. One of the challenges in adapting LLMs for MR tasks lies in the differences in data format and task requirements.

Moment retrieval tasks Lei et al. (2021) often deal with sparse training data, and the nature of the data may be different from the corpora on which LLMs were originally trained. Simply fine-tuning LLMs Liu et al. (2024) on moment retrieval data may not be sufficient to achieve optimal performance, as these models may not capture the nuances of video understanding and cross-modal reasoning. Therefore, there is a need for specialized approaches that can effectively leverage both textual and visual information for moment retrieval tasks. These approaches may involve combining LLMs with specialized models for video understanding, employing techniques such as multi-modal fusion Li et al. (2023), cross-modal attention mechanisms He et al. (2021a), and task-specific adaptations Liu et al. (2021b). By integrating these approaches, it may be possible to enhance the performance of LLMs in tasks involving videos and multiple modalities.

## 2.2 Video Moment Retrieval

MR is the task of localizing the moment relevant to the given text description. Different from the MR, HD aims to measure the clipwise importance level of the given video.

Traditional approaches like Moment context network (MCN) Anne Hendricks et al. (2017) employed a two-stage "proposal-rank" pipeline, which involved handcrafted predefined proposals. However, this approach often resulted in redundant computations due to dense proposals and a large number of negative samples. In contrast to traditional approaches, proposal-free methods like LPNet Xiao et al. (2021) directly regress start and end boundaries or probabilities without relying on predefined proposals. This direct approach typically results in faster processing times since it avoids the need for proposal generation.

Another category of methods, known as proposal-learnable methods, utilizes networks like adaptive proposal generative network (APGN) Liu et al. (2021b) to adaptively predict video segments. These methods demonstrate improved performance compared to proposal-free methods by dynamically generating proposals based on the input data. However, they may suffer from location bias which can limit their generalization ability to diverse datasets or scenarios.

Due to the similarity between MR and HD tasks based on queries, as well as the commonality in their methods involving multi-modal feature extraction and feature interaction, some studies have focused on designing various multi-task networks for joint MR and HD. In Moment-DETR Lei et al. (2021) introduced the QVHighlights dataset and modifies detection transformer (DETR) based model to handle the MR and HD jointly. Various research efforts Anne Hendricks et al. (2017); Ma et al. (2023) were put into the search for the requested moments in the video and summarizing the video highlights. However, these models still struggle with semantic of text query and relevance matching between text query and video content. They rely on separate modeling of visual and textual features, lacking deep integration between these two modalities. Recent approaches Moon et al. (2023b) have shown that deploying the cross-attention mechanism of transformer architecture is more effective to fuse the text query into the video representation. UMT Liu et al. (2022) proposed transformer based architectures to take multimodal sources, e.g., video and audio. However, it removed the moment decoder and bipartite matching, resulting in inferior performance on MR. Additionally, the potential of LLMs, a naturally powerful textual transformer decoder, remains unexplored in the MR task. MH-DETR Xu et al. (2023) incorporates a pooling operation into the encoder and integrates a cross-modality interaction module to fuse visual and query features. Despite these advancements, the existing methods still suffers from the limited data availability in the target dataset and multi-modal feature extraction and feature interaction.

## 2.3 MULTI-MODAL ALIGNMENT

Recently, there has been increasing interest in the multimodal computing field in developing contrastive losses to capture the interdependence across different modalities Wang et al. (2016). Most of these multimodal strategies Wu et al. (2013) do not explicitly align semantic information from different modalities before facilitating modality interaction. This limitation can lead to insufficient discrimination of joint features.

Adapter learning Kim et al. (2021); He et al. (2021b) for cross-modality is a specialized technique employed in machine learning to address the challenges associated with integrating information from diverse modalities. It focuses on building adaptable modules or adapters that enable a model to effectively handle information from multiple modalities. Houlsby et al. (2019) extensively studied the choices of adapter architectures and concluded that a stack of networks works well which only introduces a small amount of extra parameters to the model.

Cross-modality Yan et al. (2019) tasks involve processing and understanding data that originates from different sources or modalities, such as combining text and images. In the context of cross-modality, Lin et al. (2023b) aims to create modality-specific modules that can be easily plugged into a unified model via adapter learning. The model becomes more versatile in handling distinct types of data, allowing for improved performance on tasks that involve multiple modalities. adapter learning for cross-modality is particularly valuable in applications such as segmentation He et al. (2021a), multimodal sentiment analysis Ryumina et al. (2023).

This work is inspired from the utilization of adapter learning to enhance the flexibility and robustness of our proposed model in addressing the intricacies associated with diverse data modalities.

## 3 PROPOSED METHOD

### 3.1 PROBLEM FORMULATION

Given a query $Q$ and a video $V$ with temporal length of $L$ key frames, our main objective is to localize a target moment $\mathcal{M}$ in the given video, starting at timestamp $\mathcal{T}_\mathcal{S}$ and ending at timestamp $\mathcal{T}_\mathcal{E}$, which semantically matches the query $\mathcal{Q}$ and rank the highlight score for each clip. Our solution usually involves: 1) Data creation which leverages the ability of large language models (LLMs) to create a diverse synthetic text queries. 2) To exploit multi-modal information, we learn query and video representations so that foundation models are fine-tuned over moment-retrieval tasks via an adapter learning to model the cross-modal relationship.

### 3.2 OVERVIEW

The proposed architecture is shown in Figure 1. We augment text queries in the training set by utilizing LLMs (Sec. 3.3). Then, the framework extracts representations for videos and queries using frozen foundation models, namely SlowFast Feichtenhofer et al. (2019) and CLIP Radford et al. (2021) for videos, and CLIP for queries. Furthermore, we incorporate nested adapter to facilitate domain specific video-query alignment. The baseline MR pipeline involves initial steps of utilizing pretrained models to extract features from both the input video and query. Subsequently, cross-modal interaction is employed based on these features to derive the query relevance score for the video moment.

We adopt the transformer-based architecture from previous works, including QD DETR Moon et al. (2023b), Moment DeTR Lei et al. (2021), and CG DETR Moon et al. (2023a), for the encoder-decoder framework. Each encoder layer consists of a multi-head self-attention mechanism and a feed-forward network (FFN). In the context of Moment Retrieval and Highlight Detection (MR/HD), the encoder generates clip-wise representations enriched with query-relevant information.

To improve the diversity and quality of query formulations, we employ GPT-3.5-turbo and GPT-4-o-mini, leveraging their ability to generate varied yet semantically coherent queries in order enhance the model's ability to generalize and retrieve relevant moments more effectively. Cross-attention between video and query modalities is applied early in the encoder to ensure query-dependent video representations. These features are then used for tasks such as retrieving relevant moments (MR) and predicting clip-wise saliency scores (HD).

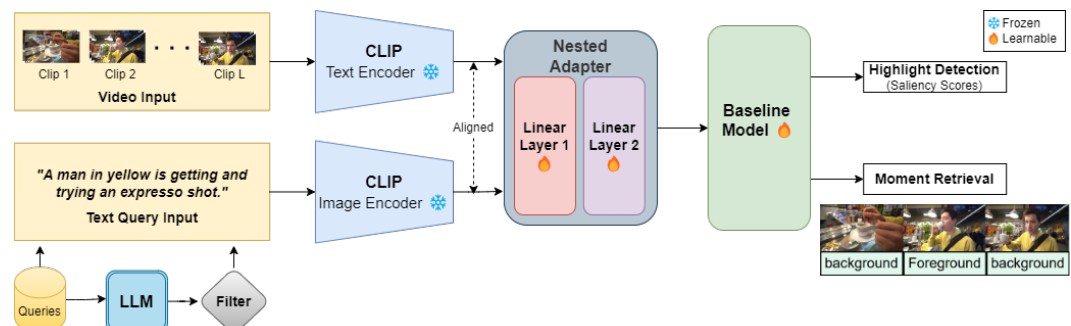

Figure 1: Figure 1: Proposed Architecture. In the Nested Adapter, the second layer is trained only for original queries, while the first layer is trained for both original and synthetic queries. Baseline models used for experiments were Moment DETR, QD DETR, and CG DETR.

## 3.3 DATA AUGMENTATION VIA LLMS

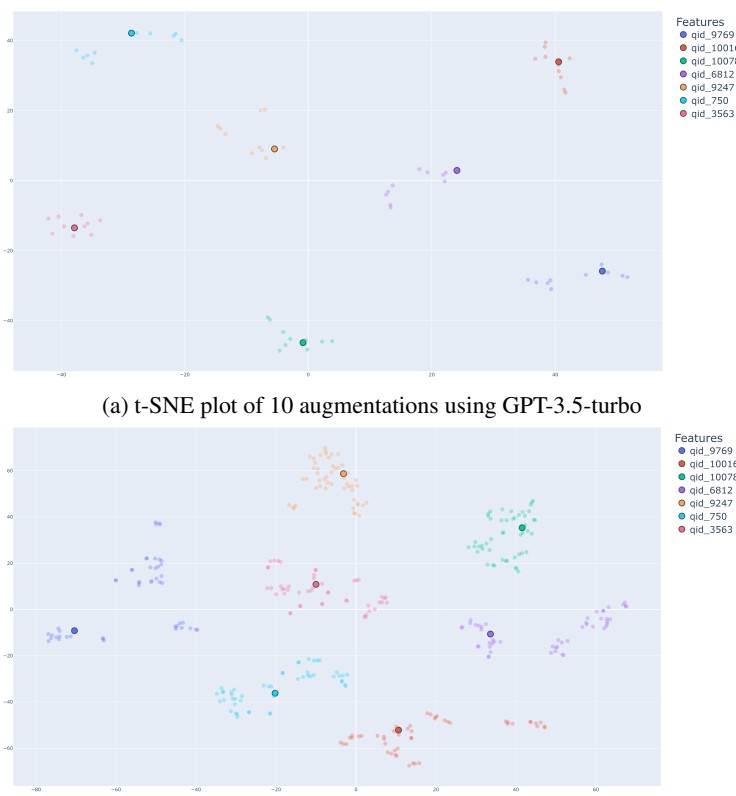

(a) t-SNE plot of 10 augmentations using GPT-3.5-turbo

(b) t-SNE plot of 50 augmentations generated using GPT-4o-mini

Figure 2: t-SNE visualizations of query augmentations generated by GPT-3.5-turbo (10 augmentations) and GPT-4o-mini models (50 augmentations). (a) and (b) show query features and its augmentations.

Data augmentation Ding et al. (2024) is a crucial technique for enhancing model performance by generating diverse training examples without requiring additional data collection. In the context of large language models (LLMs), augmentation has been particularly impactful, allowing for the creation of varied and high-quality synthetic data.

For Video MR, augmenting queries via LLMs involves generating alternative phrasings, contexts, and variations of the original queries while preserving semantic intent (see Fig: 2). This process

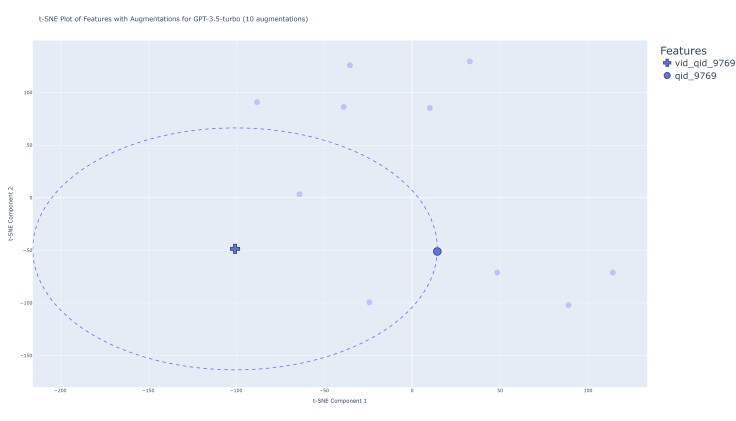

(a) t-SNE plot of 10 augmentations generated using GPT-3.5-turbo for qid-9769

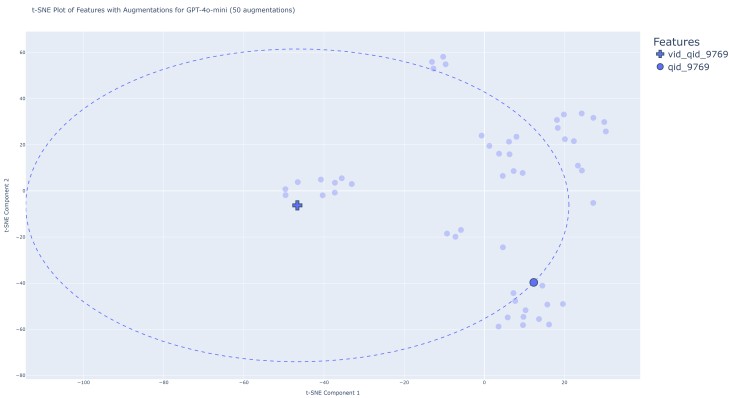

(b) t-SNE plot of 50 augmentations generated using GPT-4o-mini for qid-9769

Figure 3: t-SNE visualizations of query augmentations generated by GPT-3.5-turbo (10 augmentations) and GPT-4o-mini models (50 augmentations) (a) and (b) focus on augmentations for a particular query with reference to its corresponding video (see A.1).

includes paraphrasing, reordering words, and introducing synonyms, all of which contribute to a diverse set of queries. The enhanced dataset allows the model to generalize better to varied inputs.

The key benefits of using LLMs for query augmentation are:

**(1) Data Diversity**: Augmenting queries through paraphrasing increases the dataset's lexical diversity by employing synonyms, alternative phrasings, and varied vocabulary. This provides broader coverage of language variations and improves the model's robustness.

**(2) Correctness**: Paraphrased queries maintain grammatical correctness and clarity, ensuring that the augmented data remains syntactically sound and easy to understand.

**(3) Semantic Coherence**: Despite lexical variations, the paraphrased queries preserve the original semantic intent with slight linguistic perturbations, ensuring that augmented queries remain relevant and contextually accurate.

By leveraging LLMs for query augmentation, the model is exposed to a wider range of linguistic structures and semantic perturbations, improving its overall performance on diverse inputs.

We employed two data augmentation strategies: (1) randomly selecting $k$ paraphrased queries for each original query, and (2) using a distance-based metric to select the most semantically suitable paraphrased query. For the distance-based approach, we compute the semantic distance between

each paraphrased query $\mathbf{q}_i'$ and the relevant video clip (ground truth moment label) $\mathbf{c}_j$. The semantic distance is represented as $d(\mathbf{q}_i', \mathbf{c}_j)$, where $d(\cdot, \cdot)$ is a predefined distance function (e.g., cosine distance in the embedding space). We select paraphrased queries such that:

$$d(\mathbf{q}_i', \mathbf{c}_j) < \delta \cdot d(\mathbf{q}_i, \mathbf{c}_j)$$

where $\mathbf{q}_i$ is the original query. $\delta$ is a scaling factor, which tightens the distance for augmented queries compared to original queries. We use cosine distance to measure all distances. Both methods demonstrate improvements over the baseline, with distance-based data augmentation consistently outperforming simple random selection. This highlights the value of incorporating diverse and contextually relevant queries to enhance model performance.

### 3.4 NESTED ADAPTER

Adapter-based tuning has emerged as an effective method for enhancing pre-trained models by integrating lightweight adapter modules. This process involves fine-tuning a pre-trained model on task-specific data, enabling it to learn features relevant to new tasks. Adapters are particularly adept at combining features from different modalities, such as video frames and textual descriptions, to improve performance on tasks requiring multi-modal understanding.

Let $\mathbf{q}$ represent the original query embedding, and let $\mathbf{q}_i'$ denote a specific augmented query embedding from the set of augmentations $\{\mathbf{q}'\}$ generated by large language models (LLMs). The proposed nested adapter consists of two linear layers $\mathbf{W}_1 \in \mathbb{R}^{d \times d'}$ and $\mathbf{W}_2 \in \mathbb{R}^{d' \times d}$ each with their corresponding bias terms $\mathbf{b}_1$ and $\mathbf{b}_2$.

The forward pass through the nested adapter is described as follows:

$$\mathbf{h}_1 = f(\mathbf{W}_1 \mathbf{x} + \mathbf{b}_1) \quad \text{where} \quad \mathbf{x} \in \{\mathbf{q}, \mathbf{q}_i'\}$$

$$\mathbf{h}_2 = g(\mathbf{W}_2 \mathbf{h}_1 + \mathbf{b}_2)$$

Here, $f(\cdot)$ and $g(\cdot)$ denote non-linear activation functions (e.g., ReLU or GeLU).

During the backward pass, the weight update for the second layer occurs only for the original query:

$$\Delta \mathbf{W}_2 = -\eta \frac{\partial L}{\partial \mathbf{W}_2} \quad \text{where} \quad \mathbf{x} = \mathbf{q}$$

In contrast, the weights of the first layer are updated based on both original and augmented queries:

$$\Delta \mathbf{W}_1 = -\eta \frac{\partial L}{\partial \mathbf{W}_1} \quad \text{where} \quad \mathbf{x} \in \{\mathbf{q}, \mathbf{q}_i'\}$$

This approach ensures that gradients $\frac{\partial L}{\partial \mathbf{W}_2}$ are calculated only for the original query, preventing weight updates for augmented pairs $\mathbf{q}_i'$. This segregation of training mitigates the risk of model degradation from low-quality augmentations, enhancing robustness and generalization. The nested adapter functions as a typical adapter in the context of video processing.

## 4 EXPERIMENTS

### 4.1 DATASETS

The **QV-Highlights** dataset contains 10,310 natural language queries associated with 18,367 moments in 10,148 YouTube videos. Each video averages around 150 seconds in duration, with approximately 1.8 distinct moments per query, and each moment lasting about 24.6 seconds. The videos are categorized into three main types: **daily vlogs**, **travel vlogs**, and **news events**. Each video is annotated with (1) a human-generated query, (2) the relevant moments in the video corresponding to the query, and (3) saliency scores on a five-point scale, ranging from 'Very Good' to 'Very Bad', for all query-relevant clips. These comprehensive annotations support the development and evaluation of systems aimed at detecting relevant moments and identifying salient highlights across diverse user queries.

AS QV-Highlights final test dataset is not publicly released, we use the validation set as a proxy for test set. The dataset we utilize consists of 7,218 video-query pairs for training and 1,550 video-query samples for evaluation.

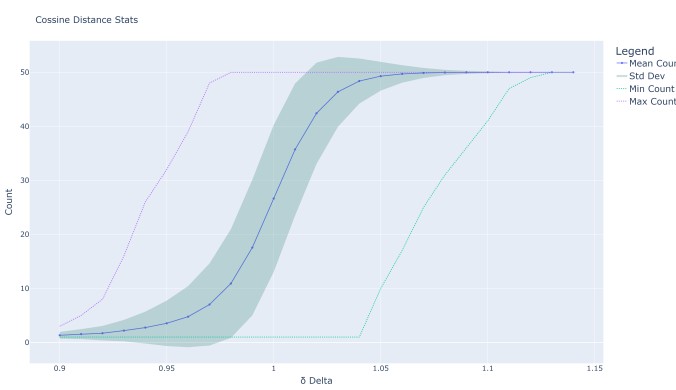

Figure 4: Count vs $\delta$ Delta Plot for Augmentations (50 for each query) generated by GPT-4o-mini

For the semantic data augmentation, we employ **GPT-3.5-turbo** and **GPT-4o-mini** to rephrase the queries. GPT-3.5-turbo generates 10 rephrased versions for each query, while GPT-4o-mini generates 50 rephrased versions per query. This augmentation was applied exclusively to the queries in the training dataset, leaving the test dataset unchanged.

## 4.2 METRICS AND EXPERIMENTAL SETTINGS

To measure the performances of QV-Highlights, we use the same evaluation metrics used in the baselines Lei et al. (2021). For each model, we report the Moment Retrieval (MR) metrics, including R1@0.5, R1@0.7, mAP, mAP@0.5, and mAP@0.75, as well as Highlight Detection (HD) metrics (mAP and Hit1). Similarly, for highlight detection assessment, we utilize mAP and HIT@1. HIT@1 is determined by the hit ratio of the highest-scored clip, where a clip is considered positive if it achieves a score of 'Very Good'.

Our model is implemented in PyTorch. For QV-Highlights, we use SlowFast Feichtenhofer et al. (2019) and CLIP Radford et al. (2021) to extract visual features, and the text encoder in CLIP for textual features. The training involves 200 epochs, a batch size of 32, and a learning rate of 1e-4. The model weights are initialized with Xavier initialization Kumar (2017). We use AdamW Loshchilov & Hutter (2017) with an initial learning rate of $1e-4$, weight decay $1e-4$ to optimize the model parameters. The models are trained with NVIDIA GeForce RTX A5000 GPU.

## 4.3 RESULTS AND DISCUSSION

Table 1: Comparison of Different Models on MR and HD Metrics - Final Results

| Baseline | Improvement | MR (Moment Retrieval) | | | | | HD (Highlight Detection) | |
|---|---|---|---|---|---|---|---|---|
| | | R1@0.5 | R1@0.7 | mAP | mAP@0.5 | mAP@0.75 | mAP | Hit1 |
| Moment DETR | None | 52.1 | 33.7 | 30.8 | 54.3 | 29.8 | 34.4 | 52.7 |
| | **Nested Adapter** ($\delta = 0.95$) *ours* | **55.9** | **36.2** | **32.3** | **56.1** | **31.2** | **36.2** | **56.4** |
| QD DETR | None | 62.4 | 45.0 | 39.9 | 62.5 | 39.9 | 38.9 | 61.4 |
| | **Nested Adapter** ($\delta = 0.95$) *ours* | **62.7** | **46.7** | **41.2** | **63.2** | **41.5** | **39.2** | **63.2** |
| CG DETR | None | 65.6 | 51.9 | 44.6 | 64.3 | 45.2 | 41.1 | 66.3 |
| | **Nested Adapter** ($\delta = 0.95$) *ours* | **66.2** | 49.6 | **44.7** | **65.0** | 44.3 | 40.1 | 65.4 |

Table 1 presents the final results comparing different baselines and our approaches on Moment Retrieval (MR) and Highlight Detection (HD) metrics. The table evaluates three baseline models—Moment DETR, QD DETR, and CG DETR—both in their standard forms and with our proposed Nested Adapter ($\delta = 0.95$). Across all baselines, the Nested Adapter consistently improves performance in both MR and HD tasks. Notably, Moment DETR shows a significant increase in R1@0.5 from 52.1 to 55.9 and in mAP from 30.8 to 32.3 for MR, while its Hit1 score for HD rises from 52.7 to 56.4. Similarly, QD DETR and CG DETR also benefit from the Nested Adapter, with QD DETR's mAP increasing from 39.9 to 41.2 and CG DETR's R1@0.5 improving from 65.6 to 66.2. These results clearly demonstrate the effectiveness of the Nested Adapter in enhancing retrieval precision and improving highlight detection capabilities across different models.

We conducted extensive experiments with Moment DETR to identify values of $\delta$ that yield optimal results. These same values were applied to QD DETR, where we observed similar improvements. However, with CG DETR, the proposed approach resulted in comparatively smaller performance gains. As part of future work, we could explore more suitable $\delta$ values for distance-based coarse tuning specific to CG DETR. For clarity, we present our results for $\delta = 0.9$, $\delta = 0.95$, and $\delta = 1.0$ to demonstrate the efficacy of our method. Interestingly, CG DETR performed better with a higher number of randomly selected augmentations compared to the other two baselines (see Table 2). Empirically, CG DETR appears less sensitive to noisy semantic perturbations, as its performance consistently improved from **1x** to **8x** augmentation, unlike the other baselines where the performance didn't steadily rise always with increasing randomly selected augmentations.

In order to assess the effectiveness of our proposed approach, we perform several ablation studies focusing on two key components: data augmentation and the integration of adapters. While the primary objective is to explore how these elements influence the overall performance of video moment retrieval (MR) and highlight detection (HD), we also consider different configurations and variations within each. The first set of ablations examines the impact of augmentation size, generated using large language models (LLMs). The second set of ablations focuses on the role of adapters, including standard and Nested Adapters, and their effectiveness at fine-tuning and coarse-tuning the model for domain alignment. These studies aim to demonstrate the contribution of each component and justify their integration into the overall system.

### 4.3.1 ABLATION STUDY ON DATA AUGMENTATION

In this experiment, we investigate the impact of different levels of data augmentation on model performance. Specifically, we use large language models (LLMs) such as GPT-3.5-turbo and GPT-4o-mini to generate rephrased queries at varying augmentation sizes (1x, 2x, 4x, and 8x) and with distance-based scaling factors $\delta = 0.9$, $\delta = 0.95$, and $\delta = 1.0$.

Table 2: Ablation study over Augmentations

| Baseline | LLM | Augmnetation Size | MR (Moment Retrieval) | | | | | HD (Highlight Detection) | |
|---|---|---|---|---|---|---|---|---|---|
| | | | R1@0.5 | R1@0.7 | mAP | mAP@0.5 | mAP@0.75 | mAP | Hit1 |
| Moment DETR | None | None | 52.1 | 33.7 | 30.8 | 54.3 | 29.8 | 34.4 | 52.7 |
| | GPT-3.5-turbo | 1x Aug | 51.7 | 32.7 | 30.4 | 53.7 | 28.7 | 34.8 | 54.5 |
| | | 2x Aug | 51.2 | 32.7 | 29.9 | 53.2 | 28.5 | 35.0 | 53.5 |
| | | 4x Aug | 53.1 | 33.5 | 31.1 | 54.9 | 29.5 | 35.1 | 53.7 |
| | | 8x Aug | 53.0 | 34.2 | 31.1 | 54.5 | 29.8 | 35.5 | 55.3 |
| | | $\delta = 0.9$ | 53.2 | 35.5 | 31.5 | 55.4 | 30.5 | 35.3 | 54.8 |
| | | $\delta = 0.95$ | **54.7** | **36.1** | 32.2 | **56.5** | 31.0 | 35.9 | 55.6 |
| | | $\delta = 1.0$ | 53.6 | 33.7 | 31.2 | 54.6 | 29.7 | 35.4 | 55.5 |
| | GPT-4o-mini | $\delta = 0.9$ | 53.5 | 34.6 | 31.9 | 55.3 | 30.5 | 35.9 | 54.9 |
| | | $\delta = 0.95$ | 54.5 | 35.9 | **32.9** | 56.1 | **31.2** | **36.2** | **56.4** |
| | | $\delta = 1.0$ | 51.4 | 33.3 | 30.3 | 53.1 | 29.0 | 35.2 | 54.1 |
| QD DETR | None | None | 62.4 | 45.0 | 39.9 | 62.5 | 39.9 | 38.9 | 61.4 |
| | GPT-3.5-turbo | 1x Aug | 61.2 | 45.9 | 40.9 | 61.4 | 41.5 | 38.7 | 61.0 |
| | | 2x Aug | 62.1 | 44.9 | 40.0 | 61.9 | 40.8 | 38.9 | 60.5 |
| | | 4x Aug | 61.1 | 45.8 | 40.5 | 61.7 | 40.8 | 39.3 | 62.6 |
| | | 8x Aug | 61.7 | 46.3 | 41.1 | 61.5 | 42.3 | 38.7 | 61.6 |
| | | $\delta = 0.9$ | 62.4 | 45.9 | 40.4 | 62.0 | 41.2 | **39.5** | 61.0 |
| | | $\delta = 0.95$ | **62.6** | **46.7** | 41.2 | **63.2** | 41.5 | 39.2 | 59.2 |
| | | $\delta = 1.0$ | 61.9 | 45.5 | 40.6 | 62.3 | 40.5 | 39.1 | 60.3 |
| | GPT-4o-mini | $\delta = 0.9$ | 61.8 | 45.5 | 40.6 | 62.3 | 40.5 | 39.1 | 61.3 |
| | | $\delta = 0.95$ | 62.5 | 46.6 | **41.4** | **63.2** | **41.7** | 39.1 | **62.4** |
| | | $\delta = 1.0$ | 61.5 | 45.9 | 40.9 | 62.1 | 40.8 | 39.0 | 60.8 |
| CG DETR | None | None | **65.6** | **51.9** | **44.6** | 64.3 | 45.2 | **41.1** | **66.3** |
| | GPT-3.5-turbo | 1x Aug | 61.9 | 47.6 | 42.5 | 62.5 | 43.7 | 39.1 | 61.5 |
| | | 2x Aug | 63.5 | 48.7 | 43.3 | 63.5 | 43.8 | 40.0 | 63.6 |
| | | 4x Aug | 65.0 | 48.9 | 43.4 | 64.2 | 43.4 | 39.8 | 63.7 |
| | | 8x Aug | 65.7 | 50 | 44.3 | **64.9** | 44.5 | 40.5 | 63.8 |
| | | $\delta = 0.9$ | 64.4 | 50.7 | 44.0 | 64.2 | 44.6 | 40.3 | 64.6 |
| | | $\delta = 0.95$ | 65.3 | 51.1 | 43.9 | 64.6 | 45.0 | 40.5 | 65.5 |
| | | $\delta = 1.0$ | 65.2 | 51.3 | 44.3 | 64.7 | **45.3** | 40.2 | 63.9 |
| | GPT-4o-mini | $\delta = 0.9$ | 64.1 | 50.0 | 43.4 | 63.7 | 43.5 | 39.9 | 65.3 |
| | | $\delta = 0.95$ | 65.5 | 51.2 | 44.0 | 64.1 | 44.0 | 40.1 | 66.2 |
| | | $\delta = 1.0$ | 64.5 | 50.5 | 43.5 | 63.5 | 43.7 | 39.8 | 65.5 |

As shown in Table 2, the results highlight how augmenting the training data improves performance across different models and configurations, with specific attention to how scaling the augmentation size for both random selection and distance-based augmentation enhance both retrieval and detection tasks. For instance, the mAP score increases consistently as we move from no augmentation to larger augmentation sizes, reflecting that the model benefits from the additional semantic diver-

sity introduced by rephrased queries. However, the gain diminishes slightly after 4x augmentation, suggesting that overly extensive augmentation may lead to diminishing returns or even noise in the training data. This highlights the challenge of balancing diversity with the risk of noisy augmentations. To address this, we employ a distance-based augmentation selection strategy, ensuring that the augmentations maintain semantic relevance while reducing noise.

For distance-based augmentation in Table 2, we observe that a moderate scaling factor ($\delta = 0.95$) consistently delivers the best performance across all baselines. Lower scaling values ($\delta = 0.9$) result in less effective domain adaptation, while higher values ($\delta = 1.0$) may overfit to the augmented data, thereby reducing the model's ability to generalize.

### 4.3.2 ABLATION STUDY ON ADAPTER TYPES

Table 3: Ablation study over Adapters

| Baseline | LLM (Augmentation) | Adapter | MR (Moment Retrieval) | | | | | HD (Highlight Detection) | |
|---|---|---|---|---|---|---|---|---|---|
| | | | R1@0.5 | R1@0.7 | mAP | mAP@0.5 | mAP@0.75 | mAP | Hit1 |
| Moment DETR | None | None | 52.1 | 33.7 | 30.8 | 54.3 | 29.8 | 34.4 | 52.7 |
| | GPT-3.5-turbo ($\delta = 0.95$) | Adapter | 53.2 | 35.5 | 31.5 | 55.4 | 30.5 | 35.3 | 54.8 |
| | | Nested Adapter | 53.5 | 34.6 | 31.9 | 55.3 | 30.5 | 35.9 | 54.9 |
| | GPT-4o-mini ($\delta = 0.95$) | Adapter | 55.0 | 35.9 | 32.2 | 56.0 | 31.0 | **36.2** | **56.4** |
| | | Nested Adapter | **55.9** | **36.2** | **32.3** | **56.1** | **31.2** | **36.2** | **56.4** |
| QD DETR | None | None | 62.4 | 45.0 | 39.9 | 62.5 | 39.9 | 38.9 | 61.4 |
| | GPT-3.5-turbo ($\delta = 0.95$) | Adapter | 61.8 | 47.1 | 41.1 | 62.4 | 41.8 | 38.9 | 62.3 |
| | | Nested Adapter | 62.6 | **47.2** | 41.6 | 62.5 | **42.7** | 39.1 | 62.9 |
| | GPT-4o-mini ($\delta = 0.95$) | Adapter | 62.4 | 46.6 | 41.3 | 62.4 | 42.0 | 39.0 | 62.3 |
| | | Nested Adapter | **63.2** | 46.7 | **42.4** | **63.2** | 41.5 | **39.2** | **63.2** |
| CG DETR | None | None | 65.6 | **51.9** | 44.6 | 64.3 | 45.2 | **41.1** | **66.3** |
| | GPT-3.5-turbo ($\delta = 0.95$) | Adapter | 65.7 | 50.0 | 44.3 | 64.9 | 44.5 | 40.5 | 63.8 |
| | | Nested Adapter | 65 | 50.4 | 44.4 | 64.4 | 44.9 | 40.4 | 64.5 |
| | GPT-4o-mini ($\delta = 0.95$) | Adapter | 65.6 | 49.9 | 43.0 | 64.4 | 44.2 | 40.7 | 64.6 |
| | | Nested Adapter | **66.2** | 49.6 | **44.7** | **65.0** | 44.3 | 40.1 | 65.4 |

We conduct an ablation study on the effect of different adapter configurations on the model's performance. In Table 3, we compare the results for models without adapters, with standard adapters, and with Nested Adapters. Our findings show that the inclusion of standard adapters though provides a moderate boost in performance over the baseline is prone to certain noisy semantic variations from augmented queries. The Nested Adapters further enhance the model's ability to generalize by allowing for *coarse-tuning* with augmented queries. Specifically, the model equipped with Nested Adapters at $\delta = 0.95$ achieves the best results, improving both moment retrieval and highlight detection metrics, confirming the utility of coarse-tuning in adapting to the diverse and complex nature of video data.

To explore the effect of different scaling factors for Nested Adapters, we performed an ablation study with scaling values $\delta = 0.9$, $0.95$, and $1.0$. As detailed in Tables 2, we observe that a moderate scaling factor ($\delta = 0.95$) consistently delivers the best performance across all baselines. Lower scaling values ($\delta = 0.9$) result in less effective domain adaptation, while higher values ($\delta = 1.0$) may overfit to the augmented data, thereby reducing the model's ability to generalize. This study demonstrates the importance of tuning the scaling factor in Nested Adapters to achieve the optimal balance between fine-tuning and coarse-tuning.

## 5 CONCLUSION

In this paper, we presented a novel approach to video moment retrieval by leveraging data augmentation and adaptive learning with learnable adapters. Our method addresses key challenges in video MR, including the domain gap, limited fine-tuning data, and the reliance on a single natural language query. By introducing diverse and semantically coherent query augmentations with LLMs, we enhanced the model's ability to generalize across varied query formulations. Furthermore, our use of distance-based augmentation selection mitigates the noise introduced by random augmentations, as shown in our ablation studies and supported by t-SNE visualizations. The integration of learnable adapters for both coarse and fine-tuning further improved the model's alignment with the target dataset, enabling it to retain discriminative power while being adaptable to diverse video content. Our proposed approach is model-agnostic and demonstrated consistent performance improvements across multiple baselines, proving its effectiveness in real-world moment retrieval tasks.

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

# A APPENDIX

## A.1 DISTANCE-BASED SELECTION OF AUGMENTATIONS

In Figure 3, the dotted circular line represents the decision boundary used to filter the rephrased augmentations. For both sets of augmentations generated, the distance between the video feature and the original query feature is used as the criterion to filter out features that are farther from the video feature. In our experiments as described in 3.3 we try to adjust the selection boundary by multiplying $\delta$ to the distance between the video feature and original query feature. The dotted line in the plots depict the case where the value of $\delta$ is 1.

In Figure 3, the selection boundary is shown using Euclidean distance for simplicity, though cosine distance is used in our experiments. Note that the augmentations selected may differ from the original experiments after t-SNE dimensionality reduction, which is intended to provide an intuitive explanation.

## A.2 PROMPT FOR DATA AUGMENTATION

To generate rephrase queries we leveraged GPT-3.5-turbo and GPT-4o-mini 10 and 50 rephrasing for each query respectively. We tried a few different prompts before eventually settling on:

```
"You are supposed to rephrase the given sentence {num_rephrasing} times
without changing the semantic meaning but omitting descriptive adjectives
or adverbs. You can also merge two words in such cases, or make complex
words more descriptive, or shorten the sentence.
The output should be numbered."
```

