# OpenReview forum: "A Semantic Data Augmentation driven Nested Adapter for Video Moment Retrieval"
_ICLR.cc/2025/Conference — ICLR 2025 Conference Withdrawn Submission_

### Official Review · Reviewer_PPHd · 2024-10-26

**Soundness:** 2
**Presentation:** 2
**Contribution:** 1
**Rating:** 3
**Confidence:** 4

**Summary:**

This paper presents a nested adaptor and a LLM-driven semantic data generation pipeline to improve video moment retrieval. The pipeline generates semantically similar queries to enrich query diversity, while the nested adapter uses both augmented and human-annotated queries for coarse-tuning and fine-tuning.

**Strengths:**

1. Using LLM-generated captions to enhance video moment retrieval is reasonable.
2. Experiments demonstrate some effectiveness.

**Weaknesses:**

1. The writing and figures require improvement.
2. The contribution is limited, with minimal improvements. Using LLMs to enhance textual semantics has been explored with more substantial gains (e.g. [1-2]). The nested adaptor is also trivial.

   [1]  ChatVTG: Video Temporal Grounding via Chat with Video Dialogue Large Language Models.

   [2]  Context-Enhanced Video Moment Retrieval with Large Language Models.

3. The experiments are insufficient, lacking results on popular datasets like Charades-STA and ActivityNet Captions.
4. Minor errors, such as "Figure 1: Figure 1:".

**Questions:**

See Weaknesses.

---

### Official Review · Reviewer_F28H · 2024-10-31

**Soundness:** 1
**Presentation:** 2
**Contribution:** 2
**Rating:** 3
**Confidence:** 5

**Summary:**

This paper targets the video moment retrieval (MR) task. In this paper, authors try to use the CLIP network to handle the task. LLMs are used for query augmentations, which can improve the query diversity. Two linear layers are used in the nested adapter module. T-SNE visualization and ablation study are designed to analyze the model performance. Some experiments based on some MR baselines are conducted.

**Strengths:**

The presented method is model-agnostic. It can be used in some baselines (e.g., Moment DETR, QD DETR, and CG DETR) to improve their performance.

LLMs are utilized in the designed method to generate various queries for data augmentation. Moreover, a filter is adopted to select the generated queries. Different LLMs are used to evaluate the model performance.

The t-SNE visualization can help reader understand the feature distribution.

Authors provide some details about the model implementation in Section 4.2.

**Weaknesses:**

Figure 1 is wrong. The text encoder and the image encoder should swap positions. The video input contains multiple clips, not images. Why the image encoder can be used to obtain the clip features?

Why authors only consider the R@1 results and ignore the R@5 results?

Why not present the retrieval visualization results and the eﬃciency results? They are very important for the video moment retrieval task.

The number of baseline methods are not enough. Only three methods (Moment DeTR (Lei et al., 2021), CG DETR (Moon et al., 2023a) and QD DETR (Moon et al., 2023b)) are used in the experiment section. Authors should compare more baseline (e.g., MH-DETR (Xu et al., 2023)).

The nested adapter is not novel since it only contains two linear layers. The linear layer is very common in many networks.

The caption of Figure 1 needs to be revised since there are two “Figure 1:”.

In Section 3.1, why the query corresponds to two different symbols in Line 189 and Line 191?

Figures 2-4 are not clear (especially lines) and the texts are two small. Authors should polish them carefully.

Some grammatical errors, such as “While current work has explored different model architectures” and “the training dataset only consist of …” in Abstract.

Some punctuation marks are incorrectly used, such as ’Very Good’ in Line 405 and “(see Fig: 2)” in Line 269.

**Questions:**

Please address the weaknesses.

---

### Official Review · Reviewer_twnk · 2024-11-03

**Soundness:** 2
**Presentation:** 2
**Contribution:** 1
**Rating:** 3
**Confidence:** 4

**Summary:**

This paper introduces a novel approach to enhance video-moment retrieval models that traditionally struggle with the pretrain-finetuning paradigm due to the data dilemma. The LLM-driven data augmentation enriches the semantic boundary, while the nested adapter leverages both augmented and human-annotated queries for effective model training. The approach addresses variability in video content and captures nuanced features more effectively.

**Strengths:**

A. The text of this article is clearly written and easy to understand.
B. This paper provides a detailed description of the key benefits of using LLMs for query augmentation.

**Weaknesses:**

A. This paper mentions "domain gap" in Line 58, but the corresponding example does not adequately demonstrate this gap; a more detailed explanation should be provided.
B. This paper discusses the differences in data format and task requirements for LLMs in MR in the related work section, but it fails to address the relationship between MLLMs and MR.
C. In Line 211, this paper mentions the use of GPT-3.5-turbo and GPT-4-o mini, but it does not explain why these two LLMs were chosen, nor does it discuss more diverse evaluation methods to assess the effectiveness of query generation.
D. The comparative methods in this paper are insufficient; additional baselines should be included for comparison.
E. This paper does not provide information on the number of trainable parameters and the speed of operation for the proposed method, raising concerns about its feasibility in practical applications.
F. There is a lack of visual results, such as visualizations of data augmentation and retrieval results.
G. The meaning of the bolded data in Table 2 is unclear; for instance, "65.6" in "CG DETR" for R1@0.5 is bolded, but "65.7" is not (GPT-3.5-turbo, 8x Aug).
H. This paper should provide an explanation for the values of R1@0.7 and mAP (None, None) in "CG DETR" in Table 2, particularly why the performance declined after data augmentation was applied.

**Questions:**

see the above weaknesses.

---

### Official Review · Reviewer_tqJW · 2024-11-04

**Soundness:** 2
**Presentation:** 1
**Contribution:** 2
**Rating:** 5
**Confidence:** 4

**Summary:**

The manuscript proposes a data augmentation process driven by a Large Language Model (LLM) to enrich the semantic boundaries of textual query capture by generating queries semantically similar to real queries. Experimental results on several baseline models show the effectiveness of the proposed approach.

**Strengths:**

1. The manuscript proposes a simple quality metric for retaining high-quality data-enhanced samples.
2. A new nested adapter is proposed, which utilizes augmented and manually annotated queries for coarse and fine-tuning the model, respectively.
3. The proposed data augmentation approach is model-independent and can be applied to different architectures, enhancing its adaptability

**Weaknesses:**

1. Explainability of Data Augmentation: while the manuscript proposes an LLM-driven approach to data enhancement, it may not adequately explain the selection criteria for the enhanced data and the specific impact on model performance. For example, it is too
simplistic to do research only on the number and proportion of generated queries. Second, for queries that may have semantic annotation errors, does this practice of generating approximate semantic sentences amplify the errors and degrade performance, and how robust is the model, although the model allows for the presence of a certain amount of noise.
2. Figure content error: In Figure 1, the ‘Text Encoder’ and ‘Image Encoder’ of the CLIP feature extractor are incorrectly drawn. This visual error may mislead the reader and negatively affect the accuracy and professionalism of the article.
3. Irregularity in the title of the figure: the title of Figure 1 repeats the phrase ‘Figure 1: Figure 1’, which is unnecessary and should be simplified to ‘Figure 1’. Figure 4 has no full stop.

**Questions:**

My question has been shown in the weaknesses above. In addition, are there any clear conclusions from Figures 3 and 4?

---

### Author Response · Authors · 2024-11-25
**Withdrawal of Submission to ICLR**

We would like to extend our heartfelt gratitude to the reviewers for their time, effort, and valuable feedback on our submission. Your constructive comments and suggestions have been instrumental in helping us identify areas for improvement.
After careful consideration, we have decided to withdraw our paper from the ICLR review process. We acknowledge that the manuscript in its current form does not fully justify our claims. However, we have addressed the points raised during the review process, and our responses are now supported by additional empirical evidence.
We are committed to refining our work based on your thoughtful feedback and will resubmit an improved version to another venue soon.
Thank you again for your insightful contributions, which have greatly enhanced the quality of our research.

---

### Note · Authors · 2024-11-25

I have read and agree with the venue's withdrawal policy on behalf of myself and my co-authors.